# Spontaneous Physical Activity in Obese Condition Favours Antitumour Immunity Leading to Decreased Tumour Growth in a Syngeneic Mouse Model of Carcinogenesis

**DOI:** 10.3390/cancers14010059

**Published:** 2021-12-23

**Authors:** Delphine Le Guennec, Marie Goepp, Marie-Chantal Farges, Stéphanie Rougé, Marie-Paule Vasson, Florence Caldefie-Chezet, Adrien Rossary

**Affiliations:** 1Human Nutrition Unit, Faculty of Pharmacy, Institut National de Recherche pour l’Agriculture, l’Alimentation et l’Environnement, Université Clermont Auvergne, F-63000 Clermont-Ferrand, France; delphine.r.le.guennec@gmail.com (D.L.G.); m-chantal.farges@uca.fr (M.-C.F.); stephanie.rouge@uca.fr (S.R.); m-paule.vasson@uca.fr (M.-P.V.); florence.caldefie-chezet@uca.fr (F.C.-C.); 2Centre for Inflammation Research, University of Edinburgh Medical School, Edinburgh EH16 4TJ, UK; gmarie@ed.ac.uk

**Keywords:** obesity, sedentary lifestyle, immunosenescence, physical activity, mammary carcinogenesis, lymphocytes T helper, perforin 1

## Abstract

**Simple Summary:**

With aging, a deterioration of the immune system, termed immunosenescence, leads to a loss of innate and adaptive immunity in terms of number of cells and functionality. This results in an imbalance between pro- and anti-tumour immune response. The aim of the study was to explore the impact of physical activity on the tissue environment in a murine model of breast carcinogenesis. In this model, spontaneous physical activity slows tumour growth by decreasing low-grade inflammation and promotes antitumour immunity.

**Abstract:**

Our goal was to evaluate the effect of spontaneous physical activity on tumour immunity during aging. Elderly (*n* = 10/group, 33 weeks) ovariectomized C57BL/6J mice fed a hyperlipidic diet were housed in standard (SE) or enriched (EE) environments. After 4 weeks, orthotopic implantation of syngeneic mammary cancer EO771 cells was performed to explore the immune phenotyping in the immune organs and the tumours, as well as the cytokines in the tumour and the plasma. EE lowered circulating myostatin, IL-6 and slowed down tumour growth. Spleen and inguinal lymph node weights reduced in relation to SE. Within the tumours, EE induced a lower content of lymphoid cells with a decrease in Th2, Treg and MDCS; and, conversely, a greater quantity of Tc and TAMs. While no change in tumour NKs cells occurred, granzyme A and B expression increased as did that of perforin 1. Spontaneous physical activity in obese conditions slowed tumour growth by decreasing low-grade inflammation, modulating immune recruitment and efficacy within the tumour.

## 1. Introduction

With aging, a deterioration of the immune system, termed immunosenescence, leads to a loss of innate and adaptive immunity in terms of number of cells and functionality [1]. The function and phenotype of immune cells, as well as the expression and activity of their receptors are diminished, which contributes to the loss of certain functions such as chemotaxis or cellular destruction. A chronic low-grade inflammation appears and the production of cytokines is deregulated, which accelerates tissue deterioration and the aging process [1]. Concerning innate immune cells, the number of natural killer cells (NKs) is not significantly affected by senescence, but their cytotoxicity is reduced due to an imbalance between the cytotoxicity and cytokine production which act synergistically [1]. For adaptive immunity cells, a decline in both the number of naive T lymphocytes (LT) and in their differentiation is observed in association with thymus involution. Combined with immunosenescence, an increase in the prevalence of autoimmune disease and cancer is observed with aging, particularly breast cancer in elderly postmenopausal women [1].

Menopause, a physiological process during aging in women, induces a loss of oestrogen secretion by the ovaries and an increase in body fat [2]. Adipose tissue becomes the primary site of oestrogen production by aromatase activity [2,3]. Menopause modifies the distribution of adipose tissue in women, favouring an accumulation in visceral adipose tissue and excess weight, with unfavourable effects not only on the cardio–metabolic system but also on carcinogenesis [2]. Menopause promotes obesity, inflammation and subsequently alters secretory profiles, inducing a loop that reinforces obesity [4,5].

Obesity is associated with an increased risk of breast cancer development after menopause, but also as a factor in poor prognosis and poor response to treatments [6]. Obesity induces a systemic endocrine dysfunction characterized by chronic inflammation. Inflammation of visceral adipose tissue induces a change in both insulin sensitivity and adipocytes lipolysis, which increases ectopic lipid deposition in liver and/or skeletal muscle [6,7]. In this regard, physical activity is a recommended noninvasive intervention strategy to reduce obesity, the risk of developing breast cancer and a better response to treatments [8].

Increased physical activity is an optimal way to maintain good health by enduring different beneficial effects. Exercise provides multiorgan benefits by reducing adiposity, improving insulin sensitivity and increasing cardiopulmonary capacities [9,10,11]. Circulating fatty acids are oxidized by active muscles, promoting normal serum levels [10]. Additionally, chronic physical activity is associated with a lower secretion of proinflammatory cytokines, such as TNF-α, leptin and MCP-1, from the muscles and the adipose tissue, which in turn, contributes to the attenuation of the systemic inflammation observed in obesity [11]. Indeed, chronic exercise has been shown to decrease inflammation and stress response through myokine secretion [11,12]. Notably, skeletal muscles release significant amounts of IL-6 into circulation during prolonged exercise, which acts as an anti-inflammatory myokine [13]. The scientific literature therefore suggests that physical activity, thanks to these anti-inflammatory and immunomodulatory effects, is beneficial against the development of postmenopausal breast cancer in obese women [8]. Thus, considering woman’s menopausal status, physical activity could reduce the risk of breast cancer up to 31% in postmenopausal women [8].

Despite this literature, few data are available on the evolution of the immune system during mammary carcinogenesis associated with physical activity in obesity. The aim of this study was to explore the impact of physical activity on the immune tissue environment in a model of breast carcinogenesis. We based our study on old and ovariectomized mice (C57BL/6J mice, 33 weeks old) fed with a high-fat diet and housed in either an enriched environment, promoting physical activity and social interaction or a standard environment, providing a close to sedentary condition. In this model, we showed that the enriched environment was able to modify the whole-body homeostasis via a modulation of the cytokine signalling without deep metabolic changes [14]. In this part of the study, the immune system was analysed in the tumour microenvironment and the immune organs (thymus, spleen, lymph nodes). Cytokines and chemokines implicated in inflammation control (IL-6, IL-4, IL-10) and immune recruitment (IL-5, IL-15, LIF, CX3CL1, IFN-γ…) were analysed in the tumour microenvironment, contralateral mammary gland, inguinal adipose tissue and gastrocnemius, and were related to immune checkpoint (PDL-1, PDL-2) and cytotoxic effectors (granzyme A and perforin 1). 

## 2. Materials and Methods

### 2.1. Animal Model 

This study was approved by the Animal Experimental Committee (Comité Régional d’Ethique sur l’Expérimentation Animale, No. 01095.02, Clermont-Ferrand, France) and carried out in accordance with the ethical guidelines. Female C57BL/6J mice (33 weeks, 29.6 ± 2.2 g) were purchased from the Charles River Laboratory (Lyon, France), housed at 22 ± 2 °C in standard laboratory conditions (12 h light and 12 h dark cycle on a reverse light cycle) with ad libitum access to diet and water and housed five per cage. After 2 weeks of acclimatization, the mice were ovariectomized and randomized into two groups (*n* = 10), in a standard environment (SE) or in an enriched environment (EE), both groups had a high-fat diet until sacrifice. The high-fat diet (4.3 kcal/g) was prepared by SAFE (SAFE, Augy, France) according to the American Institute of Nutrition 93 (AIN-93G) recommendations for laboratory rodent purified diets [15]. Diet composition is detailed in Appendix A. Enrichment of the environment was obtained by housing the mice in a larger cage (60 × 38 × 20 cm) and providing additional accessories (wheels, nests, tunnels, etc.). 

### 2.2. Body and Metabolic Mice Follow-Up 

Body weight and spontaneous food intake were measured twice a week throughout the experimental period (from week 0 to week 10). Body composition was individually measured twice during the experiment (weeks 4 and 8) by quantitative magnetic resonance imaging (MRI) using an EchoMRI 3-in-1 composition analyser (Echo Medical Systems, Houston, TX, USA). The spontaneous physical activity of a group was measured twice during the experiment, before and after tumour implantation (week 4 and 8), using a TSE System PhenoMaster/LabMaster (TSE System, Bad Homburg, Germany) [16]. The mice were placed in calorimetry cages (*n* = 5 per cage) for 4 days with free access to their diet and water (22 ± 2 °C, 12 h day light cycle). Spontaneous locomotor activity in the metabolic cages was measured by the breaking of 32 infrared laser beams that spanned each cage in the xy and yz planes. In 10 min intervals, TSE LabMaster software (version 5.0.6; TSE Systems, Inc., Chesterfield, MO, USA) recorded each time a series of laser beams were broken by ambulatory and rearing activity. The total amount of spontaneous physical activity in calorimetry cages was determined by calculating the sum of both vertical and horizontal (distance covered) positions.

### 2.3. Mammary Adenocarcinoma Cell Line and Fat Pad Implantation

C57BL/6 syngeneic EO771 mammary tumour cell (Centre for Stem Cell Research, Houston, TX, USA) is a medullary breast adenocarcinoma cell line isolated from spontaneous tumours in C57BL/6J mice [17]. The cells were cultured in complete RPMI 1640 medium (Biowest, Nouaille, France) supplemented with 10% foetal calf serum (Biowest), 100 μg/mL of streptomycin (Sigma-Aldrich, Saint Quentin Fallavier, France), 100 u/mL of penicillin (Sigma-Aldrich), 2 mM of glutamine (Sigma-Aldrich) at 37 °C in a humid atmosphere at 5% CO_2_. 

After 4 weeks of diet and environment, mammary neoplastic cells were orthotopically implanted into the fourth right mammary gland using the fat-pad technique [18]. Prior to implantation, the cells were detached with trypsin, filtered to prevent cell clumping, added to 100 μL of Matrigel (BD Matrigel™ Matrix, BD Biosciences, Bedford, MA, USA) with a density of 5 × 10^5^ cells per 100 μL and kept on ice until administration to the mice. Cell suspension (100 μL) were implanted in the fourth mammary gland.

Three times a week, the tumour size was determined by measuring the perpendicular diameter with a digital calliper. The tumour volume was calculated using the formula V = 4π/3 × (width/2)^2^ × (length/2), where width is the smaller of the two measurements. The Specific Growth Rate (SGR) was calculated as the increase in volume expressed in percentage per day (SGR = (ln(V2 − V1))/T2 − T1).

### 2.4. Sacrifice and Tissue Collection

The mice were sacrificed either when the tumour volume reached 2 cm^3^ or 38 days after implantation. The mice were sacrificed by injection of ketamine/xylazine (i.p. 100/10 mg/kg of body weight) (Sigma Aldrich) associated with cardiac puncture. After blood centrifugation (13,800× *g* for 10 min at 4 °C in heparinized tubes), plasma was collected, aliquoted and stored at −80 °C until analysis.

Several organs were harvested and weighed: inguinal adipose tissue (AT), contralateral mammary gland (MG), tumour (T), gastrocnemius and lymphoid organs (i.e., thymus, spleen, inguinal lymph nodes). The tumour density was calculated as the ratio between the mass and the volume. A piece of tumour and the lymphoid organs were prepared for fresh tissue phenotype. The other organs were frozen at −80 °C until analysis. The analyses were performed only on animals for which all tissues were harvested. 

### 2.5. Immune Cell Phenotyping by Flow Cytometry

At sacrifice, cell suspensions were obtained from the harvested immune organs and tumours using mechanical disruption in PBS 1X-BSA 0.5%. They were filtered using a 40 μm-pore filter (Falcon^®^ 40 µm Cell Strainer) and submitted to hypertonic RBC lysis. Cells were counted by flow cytometry using Flow-Count Fluorospheres (Beckman Coulter, Hialeah, FL, USA). Cell suspensions were diluted with staining buffer (PBS 1X—BSA 0.5%—EDTA 2 mM) and 1 × 10^6^ cells were deposited into wells of a U-bottom-shaped 96-well plate. Cells were surface-labelled with specific antibodies for 30 min at 4 °C, as indicated in Appendix A. Intracellular staining using anti-FoxP3-Biotin was performed according to the manufacturer’s instructions (eBioscience, San Diego, CA, USA). Dead cell exclusion was performed using 10 µg/mL propidium iodide labelling. Fluorescence was quantified with a four-colour Beckman-Coulter FC500 MPL Flow Cytometer (Beckman-Coulter, Hialeah, FL, USA). Data files were analysed with Kaluza 1.2 software (Beckman-Coulter, Hialeah, FL, USA). The FACS gating strategy is illustrated in Appendix A: compensations and controls used the FMO (Fluorescence Minus One) procedure with corresponding antibody isotypes. Due to the constraints of the methods, half of the animals were randomized and used in cytometry.

### 2.6. Frozen Tissues and Plasma Analysis

#### 2.6.1. Tissue Preparation 

Tumours, mammary glands, inguinal adipose tissue and gastrocnemius were cut in pieces before homogenization with a blender. Tissues were sonicated then frozen at −80 °C for a minimum of 10 min. After thawing, the samples were homogenized and centrifuged at 500× *g* for 5 min. The filtered (pore size 40 μm) supernatant were aliquoted and frozen until analysis at −80 °C.

Tissue and plasma protein quantities were assayed with a BCA kit (Interchim, Montluçon, France) based on the Biuret method with a microplate spectrophotometer reader at 550 nm (Multiskan FC, Thermo Scientific, Waltham, MA, USA). 

#### 2.6.2. Quantification of Biomarkers 

Cytokines and chemokines (cat. MADCYMAG-72K-05 and cat. MCYTOMAG-70K: VEGF-A, G-CSF, IFN-γ, IL-2, IL-3, IL-4, IL-5, IL-10, IL-13, IL-15, IP-10, MIG, LIF, MIP-1α, RANTES, IL-6, MCP-1, TNF-α), Matrix Metalloproteases (MMPs) (cat. MMMP3MAG-79K: MMP2, MMP3, proMMP9, MMP12) and the CD8+ panel (cat. MCD8MAG-48K: CD137, Granzyme B) were determined for both plasma and tissue using Multiplex Biomarker Immunoassay kits (Merck Milliplex, Masheim, France) according to the manufacturer’s instructions. The mean fluorescence intensity (MFI) was detected by the Multiplex plate reader for all measurements (Luminex System, Bio-Rad Laboratories, Feldkirchen, Germany) using a Luminex system, Bio-Rad Laboratories software version 4.2. Data from tissues with undetectable biomarkers were discarded. 

Total cholesterol and triglycerides were quantified with commercial kits from ABX Pentra (Horiba, Montpellier, France), according to the manufacturer’s instructions.

Plasma levels of 17 β-oestradiol were assayed with the immunoassay EIA kits according to the manufacturer’s instructions (Cayman Chemical, Ann Arbor, MI, USA) with a microplate spectrophotometer reader (Multiskan FC, Thermo Scientific, Waltham, MA, USA).

### 2.7. Total Tumour RNA Isolation and Real-Time PCR

The Total tumour RNA were isolated using Trizol^®^ reagent (Invitrogen, Saint Aubin, France) according to the manufacturer’s protocol, and quantified using a Nanodrop spectrophotometer (Nanodrop^®^2000, Thermo scientific, Waltham, MA, USA). Reverse transcription was performed on 1 μg of total RNA with the high-capacity cDNA kit (Applied Biosystems, Saint Aubin, France) using random hexamer (pdN6) primers in a thermocycler (Mastercycler^®^ gradient, Eppendorf, Montesson, France).

Quantification by qPCR was performed using SYBR^®^Green reagent according to the manufacturer’s instructions on a StepOne system (Applied Biosystems, Saint Aubin, France). All primers used are summarized in Appendix A. Each sample was assayed in triplicate. Relative quantification was obtained by the comparative CT method, based on the formula 2^−ΔΔCT^ [19]. GAPDH was used for normalizing data. Tbx21, Gata3, Foxp3, Cd8α, DX5, CD1d, perforin 1, granzyme A and B levels were expressed as the fold difference between diet groups. 

### 2.8. Statistical Analysis

Statistical analyses were performed using GraphPad Prism5 (GraphPad Software, Inc, La Jolla, CA, USA). Data are expressed as means ± standard deviation. Between-group comparisons were performed by a Mann–Whitney U test. Differences between experimental conditions in time were evaluated by a one- or two-way ANOVA as appropriate, followed by a Kruskal–Wallis or Bonferroni multiple comparison test. Level of significance was set at 0.05. Significances are indicated by different subscript letters or flagged as * *p* < 0.05, ** *p* < 0.01 and *** *p* < 0.001.

## 3. Results

Both mice groups were both fed with a high-fat diet. One group was housed in a standard environment (SE) and the other in an enriched environment (EE), favouring spontaneous physical activity and social interaction. 

### 3.1. Enriched Environment Enhanced Spontaneous Physical Activity and Myokin Secretions 

The enriched environment increased the level of physical activity, represented by the total energy expenditure divided by the minimum energy expenditure, for the EE mice compared to the SE mice (Figure 1A). This result is associated with a rise in the distance covered per day per mouse for the EE compared to the SE mice (Figure 1B). Despite the increase in spontaneous physical activity, lean mass was similar in both groups during the time window (Figure 1C) and at sacrifice, the muscle masses of the posterior legs were similar between the two groups (Figure 1D). 

As expected, spontaneous physical activity influenced the secretion of the myokines found in the plasma and the gastrocnemius. In the plasma, myostatin, irisin and interleukin-6 (IL-6) were decreased for the EE group compared to the SE group (Figure 2A, Figure 2B and Figure 2C, respectively). These changes indicated the metabolism regulation of myostatin and irisin, and the global anti-inflammatory effect of IL-6 induced by the spontaneous physical activity. In the gastrocnemius, the main muscle of the posterior leg, the anti-inflammatory interleukin-10 is significantly increased in the EE group compared to the SE group (Figure 2D), which is related to the decline in the IL-6/IL-10 ratio in the same groups (Figure 2E). Conversely, another proinflammatory cytokine, interleukin-1α, was increased in the gastrocnemius of the EE mice (Figure 2F) reflecting the increase in muscle exercise. Thus, the impact of spontaneous physical activity on the whole body was characterised.

### 3.2. Physical Activity Influences the Secretions of the Whole Body

Among the cytokines assayed (Appendix A), some presented a modified pattern due to the EE housing. The proinflammatory cytokine IL-15 (useful for lymphocytes’ and NK cells’ recruitment) disappeared significantly in all the tissues investigated: plasma, tumour, contralateral mammary gland, and inguinal adipose tissue (Table 1) for the EE group compared to the SE group. IL-15 also decreased in the gastrocnemius but not significantly (Table 1, *p* = 0.0508). A similar pattern is observed for leukaemia inhibitory factor (LIF), another proinflammatory cytokine associated with the control of immunity (Table 1). LIF disappeared significantly in all the tissues investigated except in the tumour, where its level is maintained without any differences between both groups. In all tissues, CX3CL1 (implied in carcinogenesis and metastasis) is also significantly decreased in the EE group compared with the SE group, with a nonsignificant decreasing trend in the plasma (Table 1). 

### 3.3. Enriched Environment Slowed Down Tumour Growth without an Impact on Body Weight

Despite the increase in physical activity, dietary intake was similar during the time window (Figure 3A). In correlation with the high-fat diet, mouse weight increased similarly in both groups during the protocol until the implantation of the EO771 tumour cells on day 42 (Figure 3B). Tumour development slowed down this weight gain and then induced a slight weight loss (Figure 3B). The weight of the adipose tissue grew throughout the experiment until tumour implantation, without any difference between the groups (Figure 3C). 

Even though the weight gain of adipose tissue was similar, tumour growth differed according to the housing environment of the mice. The mice in the EE had slower tumour growth compared with the mice in the SE significantly from day 18 of growth onwards (Figure 4A). In relation to tumour growth, the SE mice reached the 2mm₃ limit point faster than the EE mice. The SE mice showed reduced survival compared to the mice with spontaneous physical activity (Figure 4B).

### 3.4. Enriched Environment Modified the Tumour Immune Microenvironment

The tumour microenvironment of the EE mice had a lower global immune infiltrate in terms of cell number than the SE mice, mainly due to a decrease in lymphoid cell infiltrate (Figure 5A). More specifically, this decrease is associated with a change in the balance of pro- and anti-tumour immune cells. Antitumour immune cells, such as Th1, Tc or NK, were more present in the microenvironment of the EE group’s tumours compared to the SE group’s tumours (Figure 5B). Conversely, protumour immune cells, such as Th2 and MDCS, were less present in the EE group’s tumours compared to the SE group’s tumours (Figure 5B). A more precise analysis of the immune cell subtype showed a trend towards increased infiltration of cytotoxic T lymphocytes and of TAMs within the EE group (Figure 5C). A decrease in the regulatory T cells and the MDSCs, two immune-cell types linked to aggressiveness of the tumour, was also seen in the tumour microenvironment of the EE group compared to the SE group (Figure 5C,D). 

Th1 and Th2 decreased in the EE group (Figure 6A), with a more significant decrease in Th2. This decrease induced an increase in the Th1/Th2 ratio (Figure 6A). This significant change in the pro-/anti-inflammatory lymphocyte ratio is also found in the pro- and anti-inflammatory cytokines within the tumour microenvironment. 

IL-5 was significantly decreased in the tumours of the EE group compared to the SE group (Figure 6B), IL-4 also showed a decreasing trend (Figure 6D). For proinflammatory cytokines, a decreasing trend is also observed with IL-6 and INF-γ within the EE group tumours compared to the SE group (Figure 6C and Figure 6E, respectively). In addition, the ratio IFN-γ/IL-4 decreased in the tumour microenvironment of the EE group compared to the SE group (Figure 6F), resulting in a more anti-inflammatory cytokine environment. 

Tumour cytotoxic T lymphocytes (Tc) and T regulator lymphocytes (Treg) contents were similar between the SE and EE groups (Figure 7A). However, PDL1 and 2 expression tended to increase in the tumours in the EE group compared to the SE group (Figure 7C,D). NK content was similar between both groups but NKt presented a small and nonsignificant decrease in EE group compared to SE group (Figure 7B). Perforin 1 expression was significantly increased in the EE group (Figure 7E) and granzyme A expression showed a nonsignificant increase in the same group (Figure 7F). All these data suggest a regulated and functional immune response inside the tumour. As the immune infiltrate comes from other lymphoid organs, we checked immunity inside the thymus (maturation site of T lymphocytes), spleen and inguinal lymph nodes, the major and local immune pools near the tumour.

### 3.5. Enriched Environment Modified Immune Cell Maturation but Not Differenciation

Tumour volume was positively correlated with spleen weight (Figure 8A). Indeed, the spleen weight was significantly lower in the EE group compared to the SE group (Figure 8B). Despite this decrease, the environment did not affect the CD4/CD8 lymphocyte cell populations in the spleen (Figure 8C–E). 

Within the lymph nodes, the same results were found. Their weight was significantly lower for the EE group compared with the SE group, but the CD4/CD8 lymphocytes populations were not significantly different between the groups (Figure 9A,C,E,G). 

Whatever the group, thymus weights were not significantly modified (Figure 9B). Concerning the maturation profiles, thymus CD4+/CD8+ progenitors tended to increase in the EE group compared to the SE group. Conversely, there was a significant decrease in single CD4+ or CD8+ positive progenitors in the EE group thymus compared to the SE group (Figure 9F,H). 

## 4. Discussion

Aging results in the appearance of metabolic disturbances and inflammation that can be reduced by nontherapeutic interventional strategies such as nutritional support, mobility and physical activity [1]. However, menopause and aging promote the development of breast cancer, the most common cancer worldwide, accounting for 25% of new cancer cases in women, with around 1.7 million of newly diagnosed cases per year [20]. The risk of breast cancer development doubles with each decade after menopause and is further increased with obesity [21,22,23]. Most breast cancers are hormone-dependent [24,25].

As introduced, both adiposity and physical activity are able to modulate cytokine secretions and then the immune response [26,27]. However, it was recognized that the biological effects, for both high-fat diet [28] and physical activity [29,30], need a long time, respectively up to 8 and 12 weeks, to be observed in animal models. In our conditions, the enriched environment model allows for the development of spontaneous physical activity for our mice thanks to the activity wheel, with limited impact on metabolism [14]. Due to the short time window of the high-fat diet (less than 8 weeks at sacrifice), fat and lean body mass are not altered by exercise. In these conditions, modulations of the immune system observed should depend only on the cytokine signalling and not the metabolism component. 

In addition, plasma and tissue assays, such as on the gastrocnemius, make it possible to check the evolution of physical activity markers such as myostatin, irisin or IL-6. Myostatin is mainly produced in skeletal muscle in response to inflammation or oxidative stress [31]. Conversely, physical activity inhibits its transcription and secretion into the blood compartment [32]. In vitro, this decrease has been shown to enhance insulin sensitivity of adipose tissue and muscle. In vivo, in a mouse model, the reduction of myostatin reduces the amount of adipose tissue [32,33,34]. 

Irisin is another myokine, secreted mainly by skeletal muscle. Little known or described, its level seems to increase in plasma during physical activity [35]. However, it has recently been documented that adipose tissue also expresses and secretes irisin. In studies of obese rodents (diet-induced or genetically induced), mRNA levels in the muscle are shown to increase. Further, the fibronectin type III domain-containing protein 5 (FNDC5) induces the expression of uncoupled-C-protein 1 (UCP1) in adipocytes. The beneficial role of FNDC5 in the prevention of inflammation and obesity has been hypothesized and needs to be confirmed [35]. 

With regular physical exercise, an inhibition of inflammation has also been shown [12]. Skeletal muscles will release a significant amount of IL-6 transiently, which in turn will act as an anti-inflammatory myokine [1,13]. This mechanism promotes in turn the secretion of this anti-inflammatory cytokine and the eventual lowering of IL-6. Voluntary wheel activity is also able to decrease IL-6 and TNF-α circulation in a chemically induced breast cancer model [36]. In the various established rodent models of breast cancer, a significant decrease in IL-6 circulation is found after 20 weeks of training, leading to slower tumour growth.

These results from the literature reinforced our results. Mice housed in the enriched environment present an increased daily level of physical activity associated with a decrease in circulating myostatin. Concerning irisin, it can be hypothesized that irisin consumption is increased by adipocytes or its secretion is reduced by them. More data are needed on this new myokine. Moreover, physical activity increases the level of IL-10, an anti-inflammatory cytokine; and promotes the secretion of IL-1α [37,38]. In rodent models, the imposed physical activity reduces inflammation mediators and is associated with reduced tumorigenesis, as was also seen in our model. This supports the beneficial impact of spontaneous physical activity. The anti-inflammatory effect of regular physical activity is visible in the gastrocnemius, with an increase in IL-10 and a decrease in the IL-10/IL-6 ratio. However, the beneficial effects of physical activity on inflammation caused by cancer are not yet fully established [36]. Inflammation is related to all events involved in the development and progression of cancer [36,39]. 

Breast cancers are solid tumours involving a complex association between tumour cells, fibroblasts, adipocytes, extracellular matrix, vessels and immune cells [24,25]. Inflammation and increased secretion of inflammatory cytokines within adipose tissue alters the phenotypic profile of resident immune cells. Whereas in a lean person, regulatory immune cells such as eosinophils, type 2 macrophages, regulatory or helper 2 T lymphocytes are found in majority, in an obese person, the adipose tissue is mostly infiltrated with type 1 macrophages and helper 1 T cells [8,26]. These cells maintain tissue homeostasis through the excretion of cytokines and chemokines. In an obese person, the excretion of anti-inflammatory type 2 cytokines, such as IL-4, IL-5 or IL-10, is decreased in the profile of proinflammatory cytokines such as IL-6, LIF or IL-1 [23,27,40]. These alterations have demonstrated that obesity is linked to an increased risk of cancer and mortality. Recent data demonstrate an important role of tissue secretions in the development of breast cancer. The anti-inflammatory effects of physical activity also prevent the infiltration of monocytes into adipose tissue and promote the anti-inflammatory M2 phenotype to the proinflammatory M1 phenotype [18]. More generally, physical activity promotes cytotoxic immunity, with the increase in number and activity of macrophages as well as cytotoxic T-lymphocytes [39]. These anti-inflammatory effects result in a decrease in the number of circulating proinflammatory monocytes in favour of regulatory T lymphocytes [18]. Physical activity can boost the antitumour immune defences, slowing down the immunosenescence associated with aging [41]. 

In our model, spontaneous physical activity decreases lymphocyte infiltration in tumours but favours antitumour immune cells. Our results are consistent with the data in the literature. Increasing the LTh1/LTh2 ratio favours the anti-inflammatory population of lymphocytes. Immune efficiency is increased, as shown by the expression of PDL-1, perforin 1 and granzyme A, without necessarily modifying the quantity of immune secreting cells.

Another proinflammatory cytokine, IL-15, is secreted by skeletal muscle with an important metabolic role [21]. An inverse relationship between the amount of IL-15 and the amount of adipose tissue has been demonstrated in humans [21]. This link reinforces the importance of the dialogue between exercising muscle and fat tissue during obesity [21]. Its secretion by the muscles is increased during resistance or aerobic exercise [21]. In addition, IL-15 is involved in the innate immune response, particularly in the development and functionality of NK and lymphocytes [21]. Surprisingly, this cytokine was decreased in all investigated tissues in the EE group compared with the SE group. We hypothesize that it is consumed by cells of the immune system, given its involvement in the development and functionality of NKs and lymphocytes. The cytokine LIF was decreased in all investigated organs except within the tumour. This proinflammatory cytokine of the IL-6 family promotes tumorigenesis and metastasis formation. The tumour microenvironment may try to maintain its level, despite the overall decrease through spontaneous physical activity. Fractalkine/CX3CL1 also promotes tumour progression and the development of metastases by being a protumour inflammatory chemokine [42,43]. In our study, its reduction in all tissues reflected the above results. 

## 5. Conclusions

Spontaneous physical activity promotes antitumour immunity. In obese conditions, taken together, these results showed that spontaneous physical activity slowed tumour growth by decreasing low-grade inflammation and cytokines’ dialogues, modulating the recruitment and expression of efficient proteins of the immune system within the tumour.

New investigations into immune efficiency and its presence within the adipose tissue could be carried out in order to better understand the dialogue between the tissues during breast carcinogenesis.

## Figures and Tables

**Figure 1 cancers-14-00059-f001:**
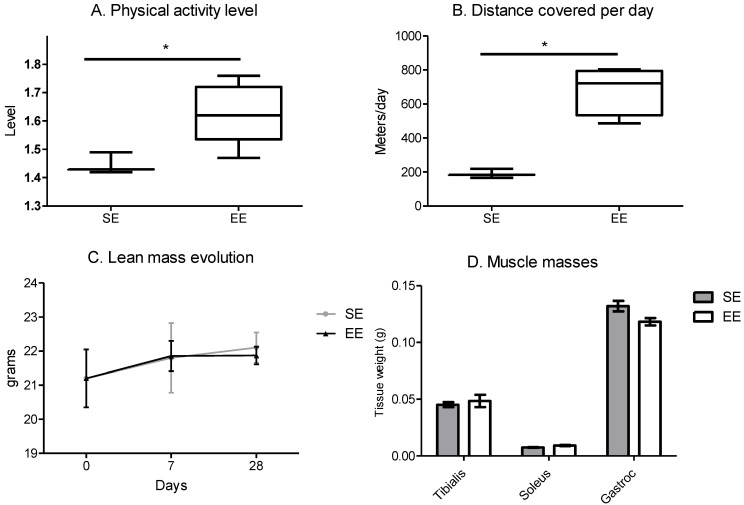
Enriched environment increases spontaneous physical activity with no change in lean mass. (**A**) level of physical activity per day per mouse, analysed by TSE system, correspond to total energy expenditure divided by the minimum energy expenditure expressed for a mouse. (means of 5 days follow-up at week 4) (**B**) distance covered per day per mouse, analysed by TSE system, results are in meters. (means of 5 days follow-up at week 4) (**C**) time course of lean mass per mouse analysed by EchoMRI. (means of 10 mice/group/days) (**D**) muscle masses of the posterior legs at sacrifice in grams (means of the two legs per 10 mice/group). Results are mean ± SEM. Data were analysed by Mann–Whitney *t*-test. Representative data are shown: * *p* < 0.05, SE vs. EE.

**Figure 2 cancers-14-00059-f002:**
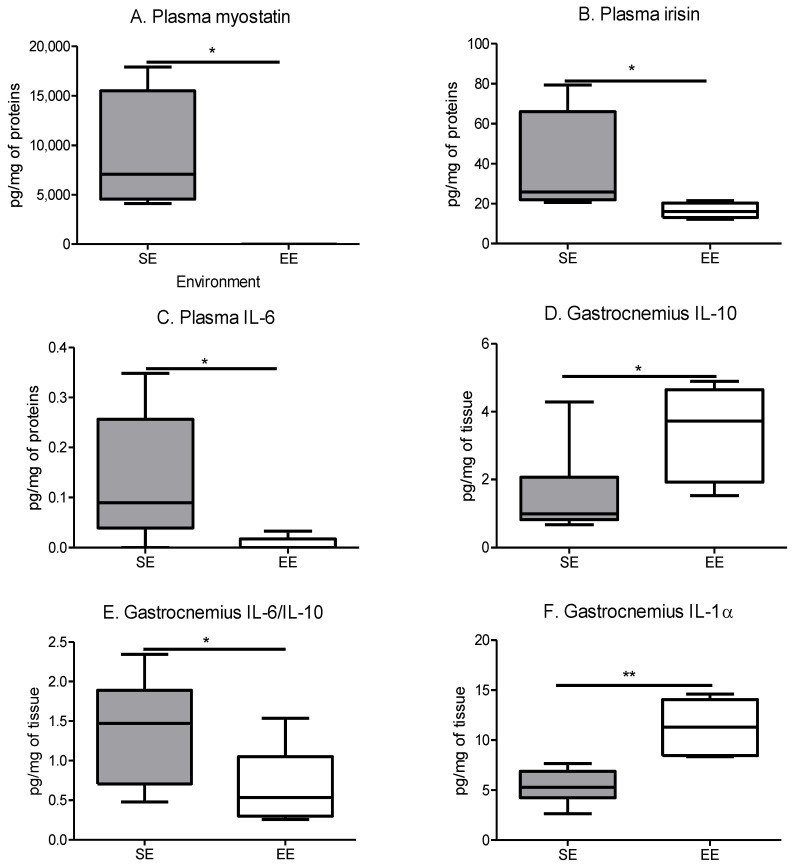
The enriched environment modifies myokine secretions like imposed physical activity. (**A**) Plasma myostatin concentration. (**B**) Plasma irisin concentration. (**C**) Plasma IL-6 concentration. (**D**) Quantity of IL-10 in gastrocnemius. (**E**) Ratio of IL-6/IL-10 in gastrocnemius. (**F**) Quantity of IL-1α in gastrocnemius. Results are in pg/mg of tissue and mean ± SEM (*n* = 5 mice muscle or plasma samples/group). Data were analysed by Mann–Whitney *t*-test. Representative data are shown: * *p* < 0.05, ** *p* < 0.01, SE vs. EE.

**Figure 3 cancers-14-00059-f003:**
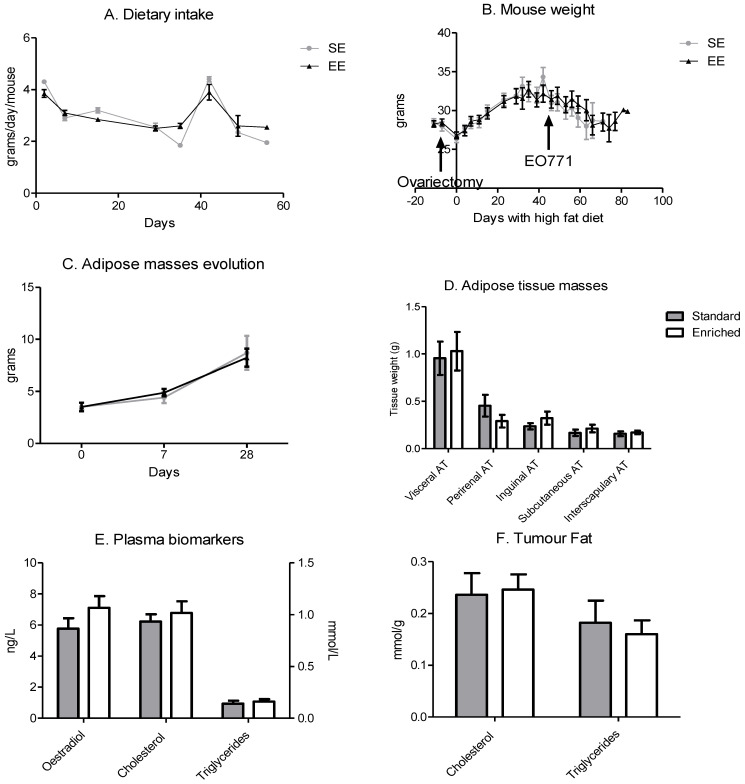
Dietary intakes are equivalent, and the high-fat diet induces an adipose mass gain. (**A**) Dietary intake per day per mouse, results are in grams per day per mouse. (means of 10 mice/group/days) (**B**) Mouse weight during experimentation. Day 0 corresponds to the beginning of the high-fat diet. The surgeries are indicated by an arrow on the graph. Results are in grams. (means of 10 mice/group/days) (**C**) Time course of adipose masses per mouse analysed by EchoMRI. (means of 10 mice/group/days) (**D**) adipose tissue masses at sacrifice in grams (means of 10 mice/group) (**E**) Quantity of oestradiol, cholesterol and triglycerides. (*n* = 5 mice plasma samples/group) (**F**) Quantity of cholesterol and triglycerides. (*n* = 5 mice tumour samples/group). Data are mean ± SEM. Data were analysed by ANOVA repeated measures.

**Figure 4 cancers-14-00059-f004:**
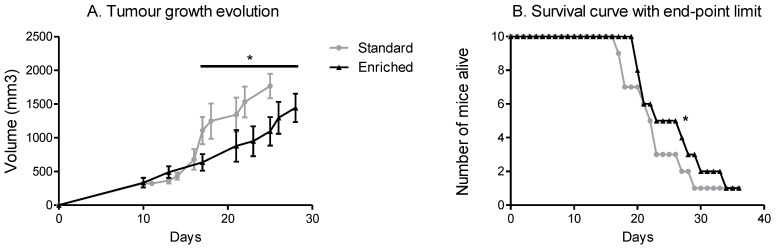
Spontaneous physical activity slows down tumour growth and increases survival. (**A**) Tumour growth evolution depending on the environment, results are in volume (mm^3^) with an individual calliper measure. (**B**). Time course of survival with end-point limit. The end-point limit is a 2 cm^3^ tumour, as required by the rules of ethics. Mean ± SEM (*n* = 10 mice/group), Data were analysed by ANOVA repeated measures or by Mantel–Cox as appropriate. Representative data are shown: * *p* < 0.05, SE vs. EE.

**Figure 5 cancers-14-00059-f005:**
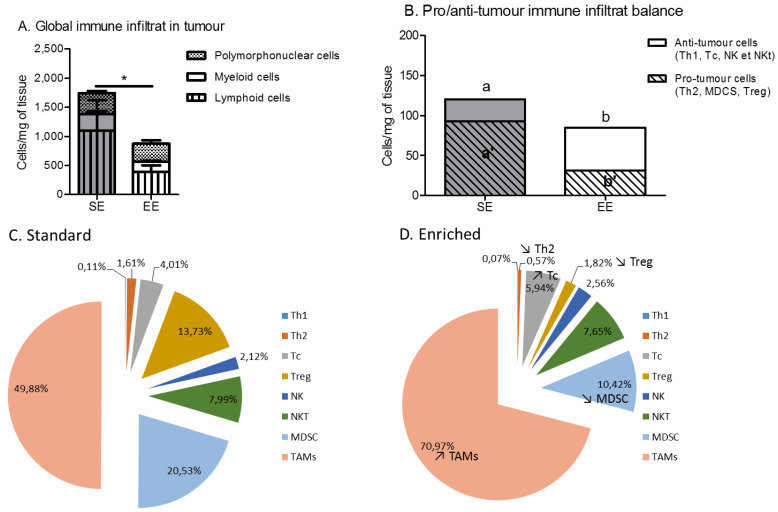
Spontaneous physical activity induces a decrease in the immune infiltrate to the benefit of anti-inflammatory and cytotoxic cells. (**A**) Global immune infiltrate in the tumours, results are in cells per mg of tumour (**B**) Pro-/anti-tumour immune infiltrate balance in tumour with results in cells per mg of tumour. (**C**) Immune phenotyping of the immune infiltrate in tumour of standard group, with results in % of immune live cells. (**D**) Immune phenotyping of the immune infiltrate in tumour of enriched group, with results in % of immune live cells. Data are expressed as mean ± SEM (*n* = 5 mice tumour phenotyping analysis/group). Data were analysed by Mann–Whitney *t*-test. Representative data are shown: * *p* < 0.05, SE vs. EE. Representative data are shown for a and b, and means without a common letter on a same line are significantly different, SE vs. EE.

**Figure 6 cancers-14-00059-f006:**
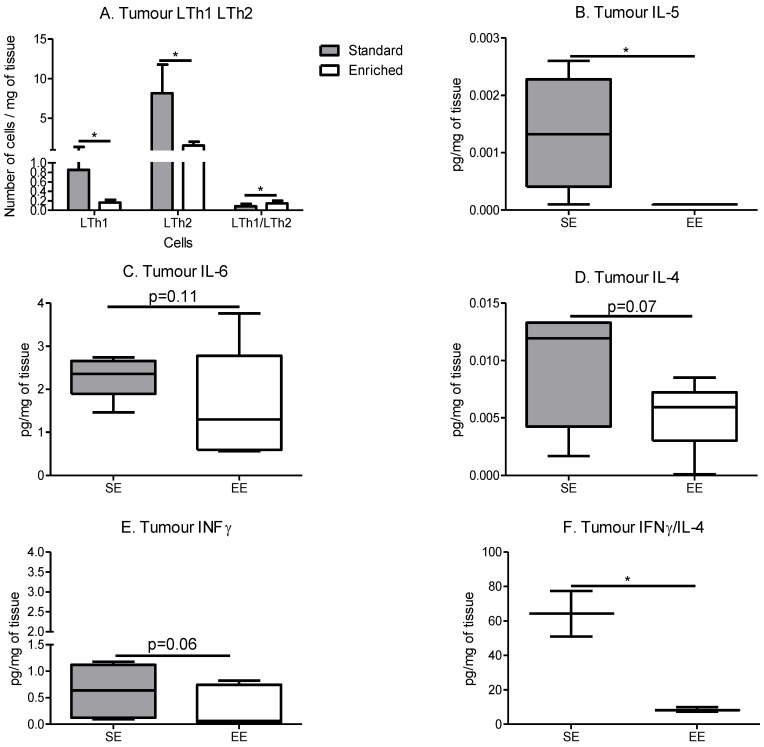
Spontaneous physical activity induces an immune anti-inflammatory microenvironment in the tumours. (**A**) Tumour lymphocytes T-helper infiltrate, results are in cells per mg of tumour. (*n* = 5 mice tumour phenotyping analysis/group) (**B**) Quantity of IL-5 in tumour, results are in pg/mg of tissue. (**C**) Quantity of IL-6 in tumour, results are in pg/mg of tissue. (**D**) Quantity of IL-4 in tumour, results are in pg/mg of tissue. (**E**) Quantity of IFN-γ in tumour, results are in pg/mg of tissue. (**F**) Ratio of IFN-γ/IL-4 in tumour, results are a ratio of IFN-γ (in pg/mg of tissue) divided by IL-4 in (in pg/mg of tissue). Cytokine determination is performed on 5 mice tumour samples/group. Data are mean ± SEM. Data were analysed by Mann–Whitney *t*-test. Representative data are shown: * *p* < 0.05, SE vs. EE.

**Figure 7 cancers-14-00059-f007:**
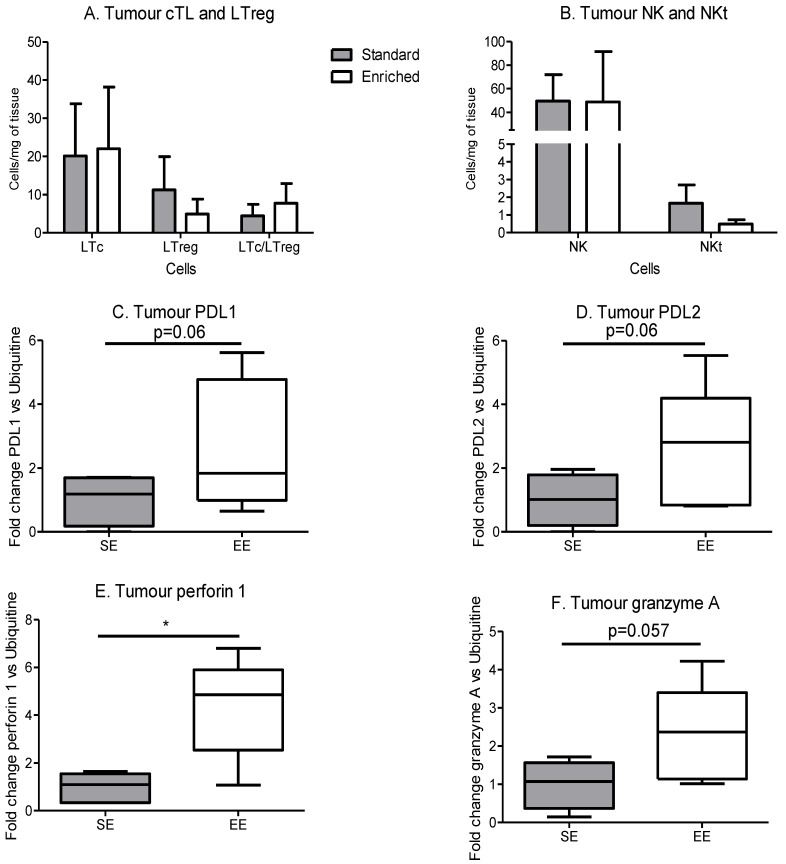
Spontaneous physical activity increases immune efficacy without changes in cTL and NKs. (**A**) Tumour lymphocytes T cytotoxic and regulator infiltrate, results are in cells per mg of tumour. (**B**) Tumour NK and NKt infiltrate, results are in cells per mg of tumour. (**C**) Expression of PDL1 in tumour microenvironment, results are in relative expression of mRNA by RT-qPCR, normalized with ubiquitin. (**D**) Expression of PDL2 in tumour microenvironment, results are in relative expression of mRNA by RT-qPCR, normalized with ubiquitin. (**E**) Expression of perforin 1 in tumour microenvironment, results are in relative expression of mRNA by RT-qPCR, normalized with ubiquitin. (**F**) Expression of granzyme A in tumour microenvironment, results are in relative expression of mRNA by RT-qPCR, normalized with ubiquitin. Phenotyping is performed on 5 mice tumour/group. mRNA expression is performed on 5 mice tumour samples/group. Data are expressed as mean ± SEM. Data were analysed by Mann–Whitney *t*-test. Representative data are shown: * *p* < 0.05, SE vs. EE.

**Figure 8 cancers-14-00059-f008:**
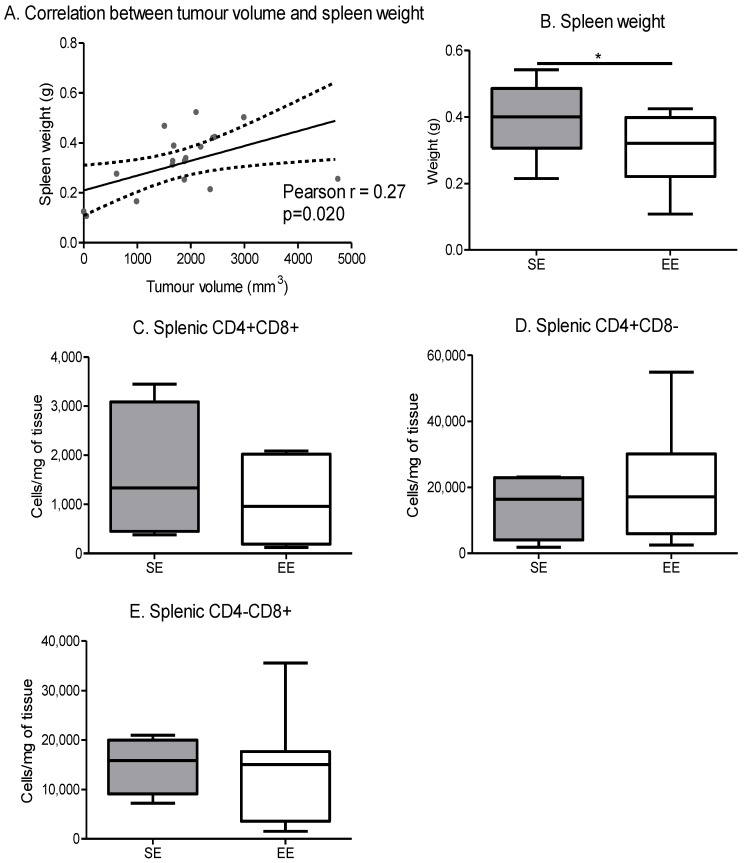
Spontaneous physical activity decreases spleen weight in correlation with tumour volume. (**A**) Correlation between tumour volume and spleen weight independent of environment, data were analysed by Pearson’s correlation test (*n* = 10 mice/group) (**B**) Spleen weight depending on the environment, results are grams (*n* = 10 mice/group) (**C**) Spleen CD4+ CD8+ maturation, results are in cells per mg of tissue. (**D**) Spleen CD4+ CD8− maturation, results are in cells per mg of tissue. (**E**) Spleen CD4− CD8+ maturation, results are in cells per mg of tissue. Phenotyping is performed on 5 mice spleen/group. Data are mean ± SEM. Data were analysed by Mann–Whitney *t*-test. Representative data are shown: * *p* ˂ 0.05, SE vs. EE.

**Figure 9 cancers-14-00059-f009:**
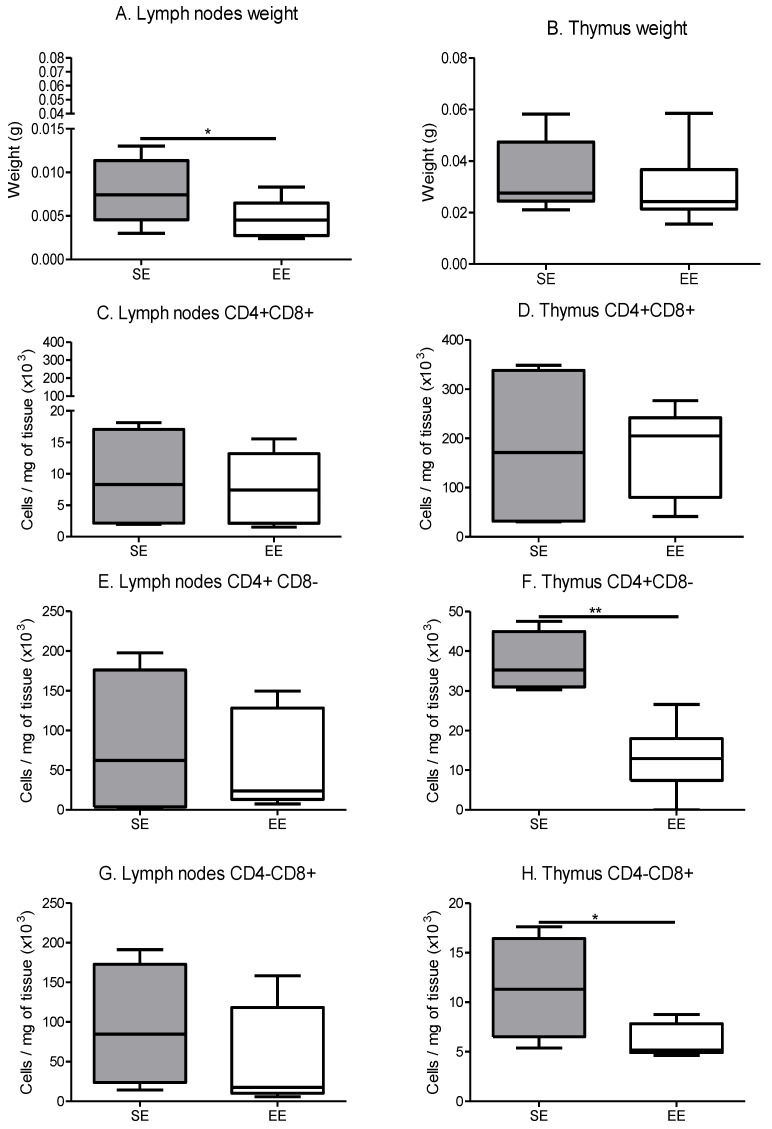
Immune lymphocytes maturation in secondary immune organs. (**A**) Lymph node weight depending on the environment, results are grams. (*n* = 10 mice/group) (**B**) Thymus weight depending on the environment, results are grams. (*n* = 10 mice/group) (**C**) Lymph node CD4+ CD8+ maturation, results are in cells per mg of tissue. (**D**) Thymus CD4+ CD8+ maturation, results are in cells per mg of tissue. (**E**) Lymph node CD4+ CD8− maturation, results are in cells per mg of tissue. (**F**) Thymus CD4+ CD8− maturation, results are in cells per mg of tissue. (**G**) Lymph node CD4− CD8+ maturation, results are in cells per mg of tissue (**H**) Thymus CD4− CD8+ maturation, results are in cells per mg of tissue. Phenotyping is performed on 5 mice tissue/group. Data are expressed as mean ± SEM. Data were analysed by Mann–Whitney *t*-test. Representative data are shown: * *p* ˂ 0.05, ** *p* < 0.01, SE vs. EE.

**Table 1 cancers-14-00059-t001:** Tissue cytokine secretions.

Samples	Cytokines (pg/mg)	Standard Environment	Enriched Environment
Plasma	IL-15	226.5 ± 49.4 a	1.00 × 10^−4^ ± 1.00 × 10^−4^ b
	LIF	6.36 ± 2.47 a	1.00 × 10^−4^ ± 1.00 × 10^−4^ b
	CX3CL1	1448 ± 404	1081 ± 253
Tumour	IL-15	0.28 ± 0.19 a	1.00 × 10^−4^ ± 1.00 × 10^−4^ b
	LIF	0.63 ± 0.17	0.67 ± 0.15
	CX3CL1	0.78 ± 0.04 a	0.55 ± 0.03 b
Contralateral mammary gland	IL-15	0.17 ± 0.17 a	1.00 × 10^−4^ ± 1.00 × 10^−4^ b
	LIF	0.002 ± 0.001 a	1.00 × 10^−4^ ± 1.00 × 10^−4^ b
	CX3CL1	0.42 ± 0.12 a	0.34 ± 0.05 b
Gastrocnemius	IL-15	1.294 ± 0.25 a	0.32 ± 0.64 b
	LIF	6.6× 10^−3^ ± 1.2 × 10^−3^ a	1.00 × 10^−4^ ± 1.00 × 10^−4^ b
	CX3CL1	0.66 ± 0.04 a	0.42 ± 0.14 b
Inguinal adipose tissue	IL-15	0.39 ± 0.15 a	1.00 × 10^−4^ ± 1.00 × 10^−4^ b
	LIF	0.020 ± 0.015 a	1.00 × 10^−4^ ± 1.00 × 10^−4^ b
	CX3CL1	0.82 ± 0.22 a	0.43 ± 0.11 b

Results are in pg/mg of proteins and mean ± SEM (*n* = 5 mice tissue or plasma samples/group). Data were analysed by two-way ANOVA for tumour-bearing mice housed in standard environment vs. tumour-bearing mice housed in enriched environment. Representative data are shown for a, b, and unlabelled (nonsignificant difference, *p* < 0.05).

## Data Availability

The datasets used and/or analysed during the current study are available from the corresponding author upon reasonable request.

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
