# Peer review of "Spontaneous Physical Activity in Obese Condition Favours Antitumour Immunity Leading to Decreased Tumour Growth in a Syngeneic Mouse Model of Carcinogenesis"

_cancers, 2021, doi:10.3390/cancers14010059_

Round 1

Reviewer 1 Report

Major concerns:

1- Figure 1: Base on the introduction and how the paper is being sold, the authors should show that beneficial effects of exercise on adipose tissue. It is not clear whether exercise exerted these beneficial effects through reduced adiposity and better control of oestrogen signaling or by other mechanisms. 

TG levels in ectopic tissues, circulating FFA, TG levels etc... should be measured 

2- Please justify the measurements in the gastrocnemius muscle, why not in other tissues, like breast, adipose, liver etc... There is no justification in the text. If for instance you want to indicate skeletal muscle, better in panels write skeletal muscle and define gastrocnemius in methods. 

3- What about the measurements of other cytokines? Where they measured with the Elisa cytokine profiling kit and showed no change? Data on all cytokines should presented (at least in supplemental). 

4- It is strange that the mice were excersising and yet no changes in adipose tissue weight were observed while calorie intake was the same? (Fig 4C). Please explain in discussion. The authors keep highlighting a role for obesity in inflammation, but so far exercise shows effects on inflammation but not on obesity in their model (TG, FFA...). There is no clear prood that exercise is leading anti-obesity effects in their model. 

5- The results section needs some more work. The authors state the results as if the reader knows already what every component is. Sometimes more explanation can help the flow of the paper (why did you check thymus weight, Why certain cytokines and not others are important, etc..). what is PLD1 why was is measured same for granzyme A etc... Not all the readers come from the same background, and the way the results section is written, makes it hard for the reader to understand and keep following the flow. 

6- Figure 4: How many times was the experiment repeated independently? this is a key experiment and the same result should be obtained at least two independent times. 

7- What about oestrogen levels following ovariectomy? 

8- Following the introduction and the way the authors justify the model, postmenoposal and female, the relationship of oestrogen with adipose tissue, and then inflammation and breast cancer, we expect to see some adipose tissue biology, oestrogen levels and changes with exercise. There is a clear dichotomy between how the topic is introduced and the choice of the model and what the results are and how they are discussed. 

Minor concerns: 

1- Line 222 -228: the text can be slightly modified to explain a bit what these changes in myokines cytokines mean. One sentence could be added to support the next step. 

2- Please revise all figure panels carefully, sometimes the titles are written in french and sometimes a mixture between english and french. (Ex. myostatin not myostatine, Cellules myleoid and many other places). 

3- Please identify clearly in legends what n means (mice, samples, replicates etc... sometimes it is not clear). 

4- Line 370: In vitro please  italicize. 

5- Figure 9A looks weird, where there data points that have been deleted? why this extension until 0.08? If there are outliers, please specify why they were deleted. 

Author Response

We thank the reviewer for the helpfull comments and reports.

We corrected the english-editing and fixed the minor mistakes in UK english.

Major concerns:

1- Figure 1: Base on the introduction and how the paper is being sold, the authors should show that beneficial effects of exercise on adipose tissue. It is not clear whether exercise exerted these beneficial effects through reduced adiposity and better control of oestrogen signaling or by other mechanisms. 

TG levels in ectopic tissues, circulating FFA, TG levels etc... should be measured 

Data concerning fat ectopic deposition and oestrogen contents are added in figure 3 with the description of energy intakes and fat mass. The different adipose tissues harvested at sacrifice are added too. The materials and methods sections was modified to include the technics employed for these determinations.

The beneficial effect of physical activity on adiposity or oestrogen signalling was not the aim of the paper, but the tumour immune response. According to the point 4 and 8 and our previous paper, we added in the beginning of discussion a part concerning the metabolic and adipose regulation in this model.

2- Please justify the measurements in the gastrocnemius muscle, why not in other tissues, like breast, adipose, liver etc... There is no justification in the text. If for instance you want to indicate skeletal muscle, better in panels write skeletal muscle and define gastrocnemius in methods. 

The gastrocnemius is the main muscle of the posterior leg. Gastrocnemius presented both glycolytic and oxidative muscle fibres what is why this muscle is a good reflect of the global metabolisms of muscle during exercise. As other muscles could have different metabolisms and response in cytokine and myokine secretions we preferred to not change the labelling to “skeletal muscle” in the different panels.

3- What about the measurements of other cytokines? Where they measured with the Elisa cytokine profiling kit and showed no change? Data on all cytokines should presented (at least in supplemental). 

A supplemental table is added (Table S4) showing cytokines and MMP contents in plasma, inguinal adipose tissue, gastrocnemius muscle and tumour. A part of these data were previously analysed in regard of the environment with several other data and published in reference 14: Le Guennec D, Hatte V, Farges M-C, Rougé S, Goepp M, Caldefie-Chezet F, et al. Modulation of inter-organ signalling in obese mice by spontaneous physical activity during mammary cancer development. Sci Rep. 29 mai 2020;10(1):8794.

4- It is strange that the mice were excersising and yet no changes in adipose tissue weight were observed while calorie intake was the same? (Fig 4C). Please explain in discussion. The authors keep highlighting a role for obesity in inflammation, but so far exercise shows effects on inflammation but not on obesity in their model (TG, FFA...). There is no clear prood that exercise is leading anti-obesity effects in their model. 

We agreed the reviewer that this point could by strange. In fact, according to the literature and the preliminary study, this point was excepted for us. We added in the beginning of the discussion some details to understand the design of the experiment, especially the time course chosen, and the consequences on the metabolic component.

5- The results section needs some more work. The authors state the results as if the reader knows already what every component is. Sometimes more explanation can help the flow of the paper (why did you check thymus weight, Why certain cytokines and not others are important, etc..). what is PLD1 why was is measured same for granzyme A etc... Not all the readers come from the same background, and the way the results section is written, makes it hard for the reader to understand and keep following the flow. 

We understand the reviewer and as recommended in the minor concerns point 1, we added some details to present the different components studied in link with the immune system and the physical activity.

6- Figure 4: How many times was the experiment repeated independently? this is a key experiment and the same result should be obtained at least two independent times. 

In agreement with the French Animal Experimental Committee and as described in the materials and methods section, the study was performed for each condition on two groups of 5 mice. Due to experimental constraints a shift over time was done between two pair of each groups. The analyses after sacrifice (cytokines, immunophenotyping) were randomly attribute between the two batch of experiment. A preliminary study was drive before to assess the feasibility and presented a similar pattern in terms of body composition and biochemistry (whole body inflammation, oestrogen, glycemia, cholesterol, triglycerides…). As immunophenotyping was not performed in this preliminary study, data are not included in this paper.

7- What about oestrogen levels following ovariectomy? 

After ovariectomy, oestrogen levels decreased from around 44 ng/l (mice fed with HFD without ovariectomy in the preliminary study) to around 6 ng/l. In this study all the mice were ovariectomised.

8- Following the introduction and the way the authors justify the model, postmenoposal and female, the relationship of oestrogen with adipose tissue, and then inflammation and breast cancer, we expect to see some adipose tissue biology, oestrogen levels and changes with exercise. There is a clear dichotomy between how the topic is introduced and the choice of the model and what the results are and how they are discussed. 

The metabolic characterisation of the model was previously published in reference 14: Le Guennec D, Hatte V, Farges M-C, Rougé S, Goepp M, Caldefie-Chezet F, et al. Modulation of inter-organ signalling in obese mice by spontaneous physical activity during mammary cancer development. Sci Rep. 29 mai 2020;10(1):8794.

This reference is added in the introduction section, and deciphered in the discussion section as follow:

“We based our study on old and ovariectomized mice (C57BL/6J mice, 33 weeks old) fed with a high fat diet and housed in either an enriched environment, promoting physical activity and social interaction or a standard environment close to sedentary condition. In this model, we showed that the enriched environment was able to modify the whole-body homeostasis via a modulation of the cytokine signalling without deep metabolic changes (14). In this part of the study, the immune system was analysed in the tumour microenvironment and the immune organs (thymus, spleen, lymph nodes). Cytokines and chemokines implied in inflammation control (IL-6, IL-4, IL-10) the immune recruitment (IL-5, IL-15, LIF, CX3CL1, IFN-γ…) were analysed in the tumour microenvironment, contralateral mammary gland, inguinal adipose tissue and gastrocnemius and related with immune check-point (PDL-1, PDL-2) and cytotoxic effectors (granzyme A and perforin 1).”

And:

“As introduced, both adiposity and physical activity are able to modulate cytokine secretions and then the immune response (26,27). But, it was recognized that the biologic effects, for both high fat diet (28) and physical activity (29,30), need long time, respectively up to 8 and 12 weeks, to be observed in animal models. In our conditions, the enriched environment model allows the development of spontaneous physical activity for our mice thanks to the activity wheel with limited impact on metabolism (14). Due to the short time window of high fat diet (less than 8 weeks at sacrifice), fat and lean body mass are not altered by exercise. In these conditions, modulations of the immune system observed should depend only of the cytokine signalling and not the metabolism component. “

Minor concerns: 

1- Line 222 -228: the text can be slightly modified to explain a bit what these changes in myokines cytokines mean. One sentence could be added to support the next step.

Some details are added to understand the changes in myokine profiles. A sentence is added to link with the next step as recommended:

“As expected, spontaneous physical activity influenced the secretion of the myokines found in the plasma and the gastrocnemius. In the plasma, myostatin, irisin and interleukin-6 (IL-6) were decreased for the EE group compared to the SE group (Figure 2A, B and C, respectively). These changes indicated the metabolism regulation for myostatin, irisin and the global anti-inflammatory effect for IL-6 of the spontaneous physical activity. In the gastrocnemius, , the main muscle of the posterior leg, the anti-inflammatory interleukin-10 is significantly increased in the EE group compared to the SE group (Figure 2D), which is related to the decline in the IL-6/IL-10 ratio in the same groups (Figure 2E). Conversely, another pro-inflammatory cytokine, interleukin-1α was increased in the gastrocnemius of the EE mice (Figure 2F) reflecting the increase of muscle exercise. Thus, the impact of spontaneous physical activity on the whole body was characterised.”

2- Please revise all figure panels carefully, sometimes the titles are written in french and sometimes a mixture between english and french. (Ex. myostatin not myostatine, Cellules myleoid and many other places). 

All figures are checked, and the corrections recommended by the reviewer are fixed.

3- Please identify clearly in legends what n means (mice, samples, replicates etc... sometimes it is not clear). 

All legends are checked, and the missing items have been specified.

4- Line 370: In vitro please  italicize. 

The correction is done.

5- Figure 9A looks weird, where there data points that have been deleted? why this extension until 0.08? If there are outliers, please specify why they were deleted. 

The figure 9 is fixed. There are not outliers on this figure. The immuno phenotyping was drove only on half of the animals as detailed in the materials and methods section. Line 163-164:

“Due to the constraints of the methods, half of the animals were randomized and used in cytometry.”

Reviewer 2 Report

This manuscript identified that spontaneous physical activity reduces chronic inflammation and inhibit tumor growth. However, more information is needed before this manuscript can be accepted for publication.

Here are a few additional major suggestions.

1 The authors should classify a, b and unlabeled (not significant difference) in the Table 1.

2 In Figure 5C, it would be better to show the total immune cells inside tumors (counts/ mg tissue or % of live cells) in the two groups. Macrophage is most abundant cell type inside the tumor. It would be better to evaluate the polarization of TAM into M1 and M2 phenotypes in cancer tissue.

3 If the authors can re-analyze the mRNA levels and show the results as fold change or scale to 100% for the SE group in Figure 7 C-F, calculate new P values and address if the changes are significant in EE in comparison to SE groups, it may get better results.

4 The authors need to provide enough information describing the gating strategies to identify different types of immune cells.

5 Whereas Th1 cells secrete the IFN-ɤ and TNF-ɑ, Th2 cells secrete IL-4, IL-5, IL-9 and IL-13. If the authors would cite more related references about these researches, which can strengthen the rationale and make the description clearer.

6 Authors need to pay more attentions to wording for the manuscript. For example, line 301 and Figure 6E, INF-y and INFɤ should be revised to IFN-ɤ. The labels of Figure 5A should be changed to polynuclear, myeloid and lymphoid. In the Figure 9, tissu should be changed to tissue. Spleen CD4+CD8+ should be changed to Splenic CD4+CD8+.

Author Response

This manuscript identified that spontaneous physical activity reduces chronic inflammation and inhibit tumor growth. However, more information is needed before this manuscript can be accepted for publication.

Here are a few additional major suggestions.

We thank the reviewer for the helpfull comments and reports.

We corrected the english-editing and fixed the minor mistakes in UK english.

1 The authors should classify a, b and unlabeled (not significant difference) in the Table 1.

The corrections recommended by the reviewer are fixed as suggested.

2 In Figure 5C, it would be better to show the total immune cells inside tumors (counts/ mg tissue or % of live cells) in the two groups. Macrophage is most abundant cell type inside the tumor. It would be better to evaluate the polarization of TAM into M1 and M2 phenotypes in cancer tissue.

Data in figure 5C already present % of living cells and the caption of figure is corrected. We agreed with the reviewer that polarization of TAM in M or M2 phenotypes inside the cancer tissue should have been more informative. Our experiment focused initially on lymphoid immune components. At the sacrifice time, the analysis set for macrophage phenotype was not implemented in the lab. That is why the data are not available for this experiment.

3 If the authors can re-analyze the mRNA levels and show the results as fold change or scale to 100% for the SE group in Figure 7 C-F, calculate new P values and address if the changes are significant in EE in comparison to SE groups, it may get better results.

The RNA levels are re-analysed and expressed as fold change. The statistical analysis of the re-analysed set of data does not change the significant. Figure 7 is corrected to present data expressed as fold change.

4 The authors need to provide enough information describing the gating strategies to identify different types of immune cells.

In agreement with the reviewer, the gating strategies are added as a supplemental figue (Figure S1).

5 Whereas Th1 cells secrete the IFN-ɤ and TNF-ɑ, Th2 cells secrete IL-4, IL-5, IL-9 and IL-13. If the authors would cite more related references about these researches, which can strengthen the rationale and make the description clearer.

6 Authors need to pay more attentions to wording for the manuscript. For example, line 301 and Figure 6E, INF-y and INFɤ should be revised to IFN-ɤ. The labels of Figure 5A should be changed to polynuclear, myeloid and lymphoid. In the Figure 9, tissu should be changed to tissue. Spleen CD4+CD8+ should be changed to Splenic CD4+CD8+.

We apologised for the wording mistakes. We corrected the english-editing and fixed the minor mistakes in UK english in the text and in the figures. The corrections recommended by the reviewer are fixed as suggested.

Round 2

Reviewer 1 Report

The manuscript has improved and is now suitable for publication